# The Effect of Herbal Medicine on Suicidal Behavior: A Protocol for Systematic Review and Meta-Analysis

**DOI:** 10.3390/healthcare11101387

**Published:** 2023-05-11

**Authors:** Chan-Young Kwon, Boram Lee

**Affiliations:** 1Department of Oriental Neuropsychiatry, College of Korean Medicine, Dong-Eui University, 52-57 Yangjeong-ro, Busanjin-gu, Busan 47227, Republic of Korea; 2KM Science Research Division, Korea Institute of Oriental Medicine, 1672, Yuseong-daero, Yuseong-gu, Daejeon 34054, Republic of Korea; qhfka9357@naver.com

**Keywords:** suicide, self-harm, herbs, herbal medicine, East Asian traditional medicine

## Abstract

Suicide is an important social and medical problem worldwide, including in countries that use traditional East Asian medicine (TEAM). Herbal medicine (HM) has been reported to be effective against several suicide-related conditions. This systematic review aimed to investigate the efficacy and safety of HM in reducing suicidal behavior including suicidal ideation, attempts, or completed suicide. We conduct a comprehensive search in 15 electronic bibliographic databases from inception to September 2022. All types of prospective clinical studies—including randomized controlled clinical trials (RCTs)—involving HM without or with routine care are included. The primary outcomes of this review are validated measures of suicidal ideation including the Beck scale for suicidal ideation. The revised Cochrane’s risk of bias tool and other tools including the ROBANS-II tool are used to assess the methodological quality of RCTs and non-RCTs, respectively. A meta-analysis is performed using RevMan 5.4 in cases of homogeneous data from controlled studies. The results of the systematic review provide high-quality evidence to determine the efficacy and safety of HM for suicidal behavior. Our findings are informative for clinicians, policymakers, and researchers, aimed at reducing suicide rates, especially in countries that use TEAM.

## 1. Introduction

As a serious public health issue worldwide, suicide is closely associated with socioeconomic disadvantage [1] and is presumably exacerbated in the context of the COVID-19 pandemic [2]. According to the data analyzed by the World Health Organization and the Global Burden of Disease Study, about 760,000 suicide-related deaths were reported in 2019, with an increase observed especially in the region of the Americas [3]. In addition, the global age-standardized rate of death by suicide in this study was 9.0 per 100,000 [3]. The COVID-19 pandemic is believed to have disproportionately adverse effects, and may have different impacts on suicide risk across different racial and socioeconomic indicators [4]. According to an interrupted time series analysis conducted in 33 countries, the overall number of suicides during the first 9 to 15 months of the COVID-19 pandemic did not change significantly compared to the pre-existing trend [5]. However, the study found exceptions where more suicides than expected were observed in certain countries or regions within countries, including lower–middle-income countries (i.e., India and Iran) [5]. In addition, a repeated cross-sectional study of eight countries on four continents during the pandemic reported high rates of suicidal ideation in adults in Hong Kong [6]. This suggests that the COVID-19 pandemic may also have a disproportionately negative impact on suicide risk.

Suicide requires a multidisciplinary and contemporary evidence-informed preventive approach [7,8,9,10]. A comprehensive and multidisciplinary approach to suicide prevention is considered an effective suicide prevention method that optimizes patient safety and considers bio-psycho-socio-cultural aspects [10]. Therefore, the expected effect and safety of available interventions to address this important issue should be reviewed from an evidence-based medicine perspective. Multidisciplinary, evidence-based suicide prevention strategies may include biologic agents according to the individual’s mental health. For example, depressive disorder is a major cause of suicide, and mental disorders are thought to contribute to suicidal behavior through complex mechanisms involving genetics, stressors, and the hypothalamic–pituitary–adrenal stress-response system [11]. In addition to depressive disorders, other mental health conditions including psychotic disorder, bipolar disorder, mental and behavioral disorders due to psychoactive substance use, anxiety disorders, and sleep disorders are also associated with a high risk of suicide [12]. The suicide rate in individuals with these psychiatric conditions is known to be about five times higher than in the general population [12]. A review article examining the biological basis of suicide and suicidal behavior reported the involvement of abnormality of serotonergic mechanism, decreased serotonin metabolites, abnormalities of receptor-linked signaling mechanisms, dysregulated hypothalamic–pituitary–adrenal axis, abnormalities of neurotrophins and their receptors, and abnormalities of neuroimmune functions [13]. Specifically, important markers of suicidal behavior include decreased levels of 5-hydroxyindoleacetic acid in cerebrospinal fluid, increased platelet 5-hydroxytryptamine 2A receptors, and abnormal hypothalamic–pituitary–adrenal axis function [13]. These biological mechanisms are the premise suggesting the possibility of using biological agents for the prevention of suicidal behavior.

Traditional East Asian medicines, including traditional Chinese medicine, Korean medicine, and Kampo medicine, are used in national medical systems in East Asian countries such as China, Korea, Taiwan, and Japan [14]. Traditional East Asian medicines have been rising in popularity for decades, with unique theories such as Yin-Yang, Five Elements and Meridian, as well as some treatments such as herbal medicine and acupuncture [15]. Traditional East Asian medicines are sometimes considered as part of complementary and alternative medicine, but traditional East Asian medicines differ in three respects: traditional East Asian medicines (1) have been regarded as a dual medical system parallel to conventional medicine; (2) include some unique treatments based on their own theory; and (3) are covered in whole or in part by medical insurance coverage in some East Asian countries [14]. Some traditional East Asian medicine modalities, such as acupuncture and herbal medicine, have been known to have potential clinical benefits for major suicidal behavior-associated psychiatric disorders, including depressive disorders [16,17], sleep disorders [18,19], and post-traumatic stress disorder [20,21]. Some herbal medicines have been shown to have anti-stress effects and have protective effects on the individual’s stress response [22]. Given that suicidal behavior is associated with a biological basis, such as a dysregulated hypothalamic–pituitary–adrenal axis [13], herbal medicines with anti-stress effects have the potential to be utilized for the prevention or improvement of suicidal behavior.

Cultural factors are an important aspect leading to consideration of traditional East Asian medicine in suicide prevention strategies. For example, suicide prevention strategies in Asian countries should take into account Asia’s unique socio-cultural context [23]. Unlike in Western countries, stigma and discrimination against people with mental illness are prevalent in Asian countries [24], which is a barrier that prevents people with mental illness at risk of suicide from accessing appropriate mental health care services in those countries, including China, Japan, Korea, Hong Kong, Singapore, and Thailand [25]. Specifically, in Asian societies, mental illness is socially less accepted and often regarded as a kind of personal weakness [25]. Also in these countries, individuals with mental illness are often considered dangerous and aggressive [25]. The stigma of mental illness may itself increase the risk of suicide by contributing to an individual’s social isolation and rejection [26]. Some individuals exposed to the stigma of mental disorders may be able to receive traditional East Asian medicine as a therapeutic alternative. This is because traditional East Asian medicine generally emphasizes mind–body–spirit connections and considers that treatment modalities of traditional East Asian medicine work on both the mind and the body [27]. Therefore, it is highly culturally relevant to investigate the effects of treatment modalities of traditional East Asian medicine, such as acupuncture and herbal medicine, on suicidal behavior in these countries. The frequent use of traditional East Asian medicine modalities in the elderly population, who are more vulnerable to suicidal behavior compared to the general population [28], also adds pressure to investigate the applicability and efficacy of the traditional East Asian medicine modality in this topic. However, the effects of traditional East Asian medicine modalities on suicidal behavior have not yet been comprehensively investigated. In particular, although clinical evidence proving the efficacy of herbal medicine for psychiatric disorders is accumulating [29], its efficacy in the context of suicide has been rarely studied.

However, the use of herbal medicine and its potential health effects is not limited to Asian countries. Individuals in Western countries who are not fully satisfied with conventional Western medicine have turned their attention to complementary and alternative medicine [30]. Therefore, some herbal medicines based on traditional East Asian medicine have been distributed to this public as herbal or dietary supplements [30]. Encouragingly, potential benefits of herbal medicine have been reported clinically for some conditions associated with suicide risks, such as depression, sleep disorders, and post-traumatic stress disorder [17,19,21,29]. For example, some herbal medicines based on traditional East Asian medicine, including *Banxia houpo* decoction, *Chaihu shugan san*, *Ganmai dazao* decoction, *Kaixin san*, *Shugan jieyu* capsules, *Sini san*, *Wuling* capsules, *Xiaoyaosan*, and *Yeuju* pill, have been widely used to treat depression for centuries, and there is preclinical and clinical evidence supporting their antidepressant effects [31,32]. Specifically, these herbal medicines are thought to exert antidepressant effects through multi-target and multi-path mechanisms, and potentially related terpenoids, flavonoids, xanthones, phenylpropanoids, and phenols have been identified [31]. Importantly, in a fluoxetine-adjunct, placebo-controlled, randomized controlled clinical trial on 18 patients with major depressive disorder, *Yeuju* pill showed a significant fast-onset antidepressant effect within 7 days [33]. However, there are issues regarding the safety of herbal medicines based on traditional East Asian medicine (e.g., *Panax ginseng C.A. Mey.*, *Glycyrrhiza glabra*, *Angelica sinensis*, and *Herba Ephedrae*) and their potential drug interactions in Western countries, which do not have strict regulations on herbal medicine compared to Asian countries [30]. Conservatively, it is estimated that one in three patients who take prescription medicine also use herbal supplements or herbal medicines [34]. The beneficial effects of herbal medicines on mental health are usually explained by their effects on central receptor activity and/or their interactions with central neurotransmitters [34]. Some popular herbal medicines used to improve mental health may have interactions with psychotropic drugs, for example, extract of *Ginkgo biloba* leaf may have effects on drug-metabolizing enzymes (e.g., inhibition of CYP450 2D6 and 3A4 [35], induction of CYP2B [36] and CYP3A4 [37]), while *Panax ginseng C.A. Mey.* has direct pharmacodynamic effects on the central nervous system, affecting GABAergic and/or serotonergic transmission [34]. These potential effects and interactions highlight the need to investigate the impacts of herbal medicines on mental health of individuals with mental illness and the pharmacokinetics and/or pharmacodynamics of psychotropic drugs from an evidence-based medicine perspective.

A systematic review is a study type that aims to identify, appraise, and synthesize all available studies on a particular topic using a predefined comprehensive and systematic methodologies including study search [38]. From the viewpoint of evidence-based medicine, systematic review plays an important role in the paradigm, as newly reported evidence can be critically evaluated and synthesized based on existing evidence and then integrated into clinical practice [39]. Suicide prevention is emphasized in a social context, and efforts to bridge the gap between evidence and policy are needed for evidence-based policy making [40]. It is therefore important to comprehensively and critically evaluate potential options for suicide prevention using a systematic review methodology [41].

The existence of publication bias for systematic reviews has been highlighted. An international survey published in 2009 found a significant number of unpublished systematic reviews [42]. In addition, even when protocols for systematic reviews have been previously registered and/or published, discrepancies between the protocols and systematic reviews are likely to exist [43]. Since a database of systematic review protocol, the Prospective Register of Systematic Reviews, was launched over a decade ago, the registration of protocols in systematic reviews has become more active, but the proportion of registered or published protocols among systematic reviews published between 2020 and 2021 was only 38% [44]. Therefore, registering a systematic review protocol, ideally receiving a peer review, may contribute to securing research transparency and reducing the risk of publication bias in systematic reviews.

This systematic review protocol describes a plan to comprehensively investigate the efficacy and safety of herbal medicine for suicidal behavior including suicidal ideation, attempts, or completed suicide in non-clinical or clinical populations.

## 2. Materials and Methods

### 2.1. Study Registration

The protocol for this review was registered in the International Prospective Register of Systematic Reviews (CRD42022334384; https://www.crd.york.ac.uk/prospero/display_record.php?ID=CRD42022334384, accessed on 10 March 2023). There were no amendments after the protocol registration. This protocol complied with the Preferred Reporting Items for Systematic reviews and Meta-Analysis Protocols 2015 statement (Appendix A) [45]. Moreover, the systematic reviews to be conducted based on this protocol adhere to the Preferred Reporting Items for Systematic reviews and Meta-Analysis 2020 statement [46].

### 2.2. Data Sources

In this systematic review, a comprehensive search is conducted in 15 electronic bibliographic databases: three Chinese electronic databases (i.e., China National Knowledge Infrastructure, VIP Chinese Science and Technology Periodicals, and Wanfang data), one Japanese electronic database (i.e., Citation Information by National Institute of Informatics), five Korean electronic databases (i.e., Research Information Sharing Service, Korea Citation Index, Oriental Medicine Advanced Searching Integrated System, Koreanstudies Information Service System, and Korean Medical database), and six English electronic databases (i.e., MEDLINE via PubMed, Excerpta Medica Database via Elsevier, Allied and Complementary Medicine Database via Elton B. Stephens Company interface, the Cochrane Central Register of Controlled Trials, PsycARTICLES via ProQuest, and Cumulative Index to Nursing and Allied Health Literature via Elton B. Stephens Company interface). Furthermore, reference lists of relevant review articles are reviewed to search for potentially missing literature or grey literature. Moreover, manual searches on clinical trial registries (e.g., International Clinical Trials Registry Platform) and Google Scholar are conducted. The search terms include ‘suicide’, ‘self-harm’, ‘self-poisoning’, ‘herbal medicine’, ‘Oriental medicine’, ‘traditional Chinese medicine’, ‘Korean medicine’, and ‘Kampo medicine’. These terms are modified to suit the language of each database. The search strategy to be used in each database is identified in Appendix A.

### 2.3. Inclusion and Exclusion Criteria

#### 2.3.1. Types of Studies

Randomized controlled clinical trials are considered the gold standard for examining the efficacy and safety of interventions. However, because suicide events are rare, they have been under-studied in interventional studies, and randomized controlled clinical trials of suicidal behaviors are likely to have small sample sizes in general to draw valid inferences [47]. Considering these limitations in intervention studies of suicide, this systematic review includes all types of intervention studies. These include randomized controlled clinical trials, non-randomized controlled clinical trials, and pre–post studies (also called before–after studies). However, observational studies such as retrospective chart reviews, case series, and case reports are excluded. Animal experiments and literature studies are also excluded.

#### 2.3.2. Types of Participants

There are no restrictions on the types of participants. In other words, this systematic review includes both clinical and non-clinical populations, and participants without or with suicidal behaviors are included. This is based on the fact that suicidal behavior is difficult to study compared to other diagnosed diseases [47], and that suicidal behavior can occur not infrequently even in non-clinical populations [48,49]. Participants without suicidal behaviors for whom suicidal behavior-related outcomes are not reported are excluded. In the analysis, the clinical and non-clinical populations are strictly separated.

#### 2.3.3. Types of Interventions

As an intervention of interest, oral herbal medicine based on traditional East Asian medicine (i.e., traditional Chinese medicine, Korean medicine, and Kampo medicine) is allowed. In this case, both oral herbal medicines used as monotherapy or adjuvant therapy are allowed. However, non-oral herbal medicines such as intravenous administration, intramuscular administration (e.g., herbal injection), herbal ointment and herbal medicine retention enema are excluded.

#### 2.3.4. Types of Comparators

For controlled studies such as randomized controlled clinical trials and non-randomized controlled clinical trials, there are no restrictions on the types of comparators. In other words, sham control, wait list, no treatment, and active comparators are all allowed as control conditions. The active comparators for suicidal behavior include the following interventions: antidepressants, ketamine, cognitive behavior therapy, dialectical behavior therapy, and other suicide prevention programs [41,50].

#### 2.3.5. Types of Outcomes

The primary outcome of this review is any validated measure of suicidal ideation including the Beck scale for suicidal ideation [51], the Depressive Symptom Index Suicidality Subscale, the Suicidal Behaviors Questionnaire, the Suicidal Ideation Attributes Scale, and the Adult Suicidal Ideation Questionnaire [52]. In addition, as secondary outcomes, any other validated or non-validated measures of suicidal behaviors (i.e., suicidal ideation, attempts, or the complete suicide act) are allowed. Moreover, some assessment scales of depression, including suicidal thoughts as its subscale (e.g., the Hamilton Depression Rating Scale [53]), are considered as secondary outcomes. Adverse events that occurred during the intervention period are considered as secondary outcomes.

### 2.4. Study Selection

The study selection process is conducted following two steps. In the first step, duplicates among the initially searched documents are removed. Then, the titles and abstracts of the documents are reviewed to assess their potential relevance. Documents that are evaluated as potentially relevant or whose evaluation is unclear are forwarded to the second step. The full texts of the documents is then reviewed in depth in the second step. The reasons for the exclusion of any studies during this step are described. Finally, the study selection process is presented in the Preferred Reporting Items for Systematic reviews and Meta-Analyses flow diagram (Figure 1). This study selection process is conducted by two researchers (B Lee and CY Kwon) independently, and any disagreement is resolved by their discussion.

### 2.5. Data Extraction

The following data are extracted from the included studies using a pre-developed and pilot-tested extraction form: name of the first author, publication year of the document, country in which the study was conducted, sample size and dropout, demographic information clinical conditions of the participants, duration and details of treatment intervention and control intervention, types of outcome measures, results for each outcome measure, adverse events, the authors’ conclusions and information for the risk of bias assessment of the study. The data extraction form is in an Excel format, and Microsoft Excel (Microsoft, Redmond, WA, USA) is used. Two researchers (B Lee and CY Kwon) conduct this extraction process independently, and any discrepancies are resolved by their discussion. During the extraction process, if missing data or ambiguous data are found, the corresponding author of the included study is contacted by e-mail to request data.

### 2.6. Quality Assessment

Since this systematic review allows three types of study designs, quality assessment tools for each design are used. The revised Cochrane risk of bias tool developed by the Cochrane group is used to assess the methodological quality (i.e., risk of bias) of randomized controlled clinical trials [54]. This tool assesses the risk of bias in randomized controlled clinical trials in five domains: bias associated with (1) the randomization process, (2) deviations from intended interventions, (3) missing outcome data, (4) with the measurement of the outcome, and (5) the selection of the reported result [54]. Each domain is evaluated with a ‘high risk of bias’, ‘some concerns’, or a ‘low risk of bias’, and the researchers follow the official guidance issued by the development group [54].

The Risk of Bias Assessment Tool for Nonrandomized Studies is used to assess the methodological quality of non-randomized controlled clinical trials [55]. This tool assesses the methodological quality of non-randomized controlled clinical studies in the following six domains: (1) the selection of participants (i.e., selection bias caused by the inadequate selection of participants), (2) confounding variables (i.e., selection bias caused by the inadequate confirmation and consideration of confounding variable), (3) the measurement of exposure (i.e., performance bias caused by the inadequate measurement of exposure), (4) the blinding of the outcome assessments (i.e., detection bias caused by the inadequate blinding of outcome assessments), (5) incomplete outcome data (i.e., attrition bias caused by the inadequate handling of incomplete outcome data), and (6) selective outcome reporting (i.e., reporting bias caused by the selective reporting of outcomes) [55]. Each domain is evaluated with a ‘high risk of bias’, ‘unclear’, or ‘low risk of bias’. This tool has proven to be of moderate reliability and promising validity in assessing the methodological quality of non-randomized controlled clinical trials [55].

The methodological quality of before–after studies is assessed using the Quality Assessment Tools by the National Heart, Lung, and Blood Institute (NHLBI). This tool proposed by the NHLBI enables the quality evaluation of a before–after study (or pre–post study) through the evaluation of the 12 items, described as ‘no’, ‘yes’, ‘not applicable’, ‘not reported’ or ‘cannot determine’ [56].

The quality assessment process is conducted by two researchers (B Lee and CY Kwon) independently, and any disagreements are resolved by their discussion.

### 2.7. Data Synthesis and Data Analysis

A qualitative analysis is conducted on all included studies, during which the composition of herbal medicine and suicidal behavior-related outcomes is mainly analyzed. If the same continuous or dichotomous variables are reported in two or more controlled studies (i.e., randomized and non-randomized controlled clinical trials), a meta-analysis is conducted using the software Review Manager 5.4 (the Cochrane Collaboration, London, UK). For continuous variables, the pooled data are presented as mean differences and their 95% confidence intervals. For dichotomous variables, the pooled data are presented as risk ratios and their 95% confidence intervals. In the meta-analysis, statistical heterogeneity is assessed using the I^2^ statistic and the χ^2^ test; I^2^ values greater than 50% are considered to represent substantial heterogeneity [57]. Since this systematic review does not place any restrictions on the characteristics of the populations to be included, considerable clinical heterogeneity of the included studies is expected; therefore, a random-effect model is used for the meta-analysis. However, the cause of heterogeneity is explored by conducting subgroup analysis according to the type of population (non-clinical or clinical) and treatment duration. Moreover, we evaluate the robustness of the meta-analysis by conducting a sensitivity analysis by removing studies with a high risk of bias and numerically distant outliers.

### 2.8. Evaluation of Evidence Certainty

For the evaluation of the level of evidence, the Grading of Recommendations Assessment, Development, and Evaluation approach is used [58]. Based on the approach, the level of evidence is evaluated in terms of risk of bias, inconsistency, indirectness, imprecision, and other considerations [58]. Each level of evidence is classified as ‘high’, ‘moderate’, ‘low’ or ‘very low’ quality. This approach is a clear and comprehensive methodology for assessing and summarizing the quality of evidence supporting recommendations [59]. The certainty of the evidence obtained through this approach can be useful not only for patients and clinicians, but also for policy makers [59]. The evaluation of evidence certainty is conducted by two researchers (B Lee and CY Kwon) independently, and any disagreements are resolved by their discussion.

### 2.9. Assessment of Publication Bias

If 10 or more studies are included in the outcomes for which the meta-analysis is conducted, the publication bias is visually evaluated using a funnel plot. An asymmetry of funnel plot would be interpreted as the presence of publication bias and/or potential small study effect [60].

### 2.10. Ethics and Dissemination

No ethical approval is required as this systematic review protocol does not involve patient recruitment or personal data collection.

## 3. Discussion

### 3.1. Suicide as a Global Health Problem

Suicide is a global health problem, and has intensified in the context of the COVID-19 pandemic [1,2,5]. In addition, the long-term effects of the COVID-19 pandemic are considered a threat to an individual’s mental health, including a potential increase in manifestations of suicidal behavior [61]. According to a study conducted in the Australian workforce, suicidal and nonfatal suicidal behaviors carry significant socioeconomic costs [62]. In this study, the economic cost of suicide and non-fatal suicidal behavior reached AUD 6.7 billion in 2014, calling for more action to reduce this burden [62]. In Japan, the cost of depression-related suicide was estimated at USD 2.5 billion in 2008 [63]. The significant socioeconomic burden associated with suicide attempts is related to the use of health services to treat the injury, the psychological and social impact of the behavior on the individual and their associates, and the long-term disability resulting from the injury [64]. Therefore, suicide prevention and measures must be a priority for healthcare professionals and policy makers worldwide today.

### 3.2. The Value of Investigating Herbal Medicine in the Context of Suicidal Behavior

#### 3.2.1. Its Potentially Beneficial Effects on Health Conditions Associated with Suicide Risk

A recent systematic review of depression, a representative mental illness associated with suicide risk [11], concluded the potential benefits of herbal medicine based on traditional East Asian medicine [17]. In addition to mental illness, the type, severity, and number of some physical conditions were significantly associated with an individual’s suicidal behavior [65,66]. These physical conditions include traumatic brain injury, sleep disorders, human immunodeficiency virus/acquired immunodeficiency syndrome, migraine, and chronic obstructive pulmonary disorder [65]. Herbal medicine based on traditional East Asian medicine has recently attracted attention as a new neuroprotective agent for traumatic brain injury [67]. Herbal medicine has also been reported to be clinically associated with improvement in primary insomnia [68], migraine [69], and chronic obstructive pulmonary disorder [70]. The associations of herbal medicine with improvements in several health conditions associated with suicide risk indicate that the possibility of this intervention potentially being included as part of a comprehensive suicide prevention strategy deserves further investigation.

#### 3.2.2. Complement the Limitations of Existing Biologics

The importance of investigating herbal medicines for suicidal behavior treatment is related to the limitations of biologics in conventional medicine. Specifically, the balance of benefits and harms in the relationship between the use of antidepressants and suicide risk has been controversial [71]. For example, a recent systematic review analyzed 27 observational studies and found that exposure to generation antidepressants, excluding selective serotonin reuptake inhibitors, was associated with a higher risk of suicide in the adult population (in depression: risk ratio = 1.29, 95% confidence interval = 1.06 to 1.57; in all indications: risk ratio = 1.45, 95% confidence interval = 1.23 to 1.70) [72]. Moreover, another recent systematic review analyzing 17 observational studies found that the use of selective serotonin reuptake inhibitors was associated with a higher risk of suicide in a pediatric and adolescent population (risk ratio = 1.28, 95% confidence interval = 1.09 to 1.51) [73]. The effect of antidepressants on the prevention of suicidal behavior in the elderly population is inconclusive [74]. In the controversial relationship between use of antidepressants and suicide risk, the harms may be related to antidepressant-induced suicidality and delayed effects of antidepressants [75,76].

On the other hand, among adverse reactions caused by herbal medicine, there is no case of increased risk of suicidal behavior [77], and the fast-onset antidepressant action of some herbal medicine is attracting attention. For example, in a randomized placebo-controlled, pilot trial on 18 patients with major depressive disorder, compared to the group using placebo plus fluoxetine, a significant reduction in depressive symptoms was found in the group using *Yeuju* pill plus fluoxetine within 7 days of treatment (*p* < 0.05) [33]. In addition, the underlying mechanism of the fast-onset antidepressant action seemed to be related to the increase in serum brain-derived neurotrophic factor (r = 0.721, *p* = 0.028) [33]. *Chaihu jia longgu muli* decoction and *Xiaochaihu* decoction, which are popular herbal medicines used for the treatment of mental disorders including depressive disorders, are also thought to exert immediate and persistent antidepressant effects through enhancement of brain-derived neurotrophic factor expression in the hippocampus [78]. Therefore, compared to conventional biological agents (e.g., antidepressants) for suicidal behavior, herbal medicine may have potential advantages in terms of benefits and harms ratio, but this should be confirmed from the perspective of evidence-based medicine, and is the reason for the need for a systematic review.

#### 3.2.3. Potential Underlying Biological Mechanisms of Herbal Medicine for Suicide Risk

It is known that mental disorders may contribute to suicidal behavior through complex mechanisms such as the hypothalamic–pituitary–adrenal axis [11], and there is evidence that herbal medicine affects shared mechanisms in terms of mental health improvement [79]. Dysregulation of the hypothalamic–pituitary–adrenal axis causes synaptic dysfunction and neuronal atrophy, and is associated with depressive behavior and even suicidal behavior [80]. A recent systematic review reviewed 36 relevant studies to investigate the role of the hypothalamic–pituitary–adrenal axis in suicide risk [80]. As a result, it was found that dysregulation of this axis is not only related to various pathophysiological processes associated to mental illness, but also to suicide risk independently of the presence of mental illness [80]. In particular, the authors emphasized that hyperactivity of the hypothalamic–pituitary–adrenal axis is an important risk factor as well as a potential therapeutic target for suicide risk [80]. Preclinical studies have reported that several herbal medicines or their constituents can improve dysregulation of the hypothalamic–pituitary–adrenal axis [79]. A review article reported that constituents in herbal medicine that are thought to exert antidepressant-like activity through action on hormone receptors on this axis include quercetin, puerarin, pseudo-hypericin, baicalin, and *Tribulus terrestris* saponins in animal models of depression [79].

In addition, reactivity to stress (i.e., resilience) can be suggested as one of the mechanisms explaining the potential effect of herbal medicine on suicidal behavior. Some herbal medicines, both in the East and in the West, have been used for the purpose of promoting individual homeostasis and reducing responsiveness to stress [22]. In Western countries, herbs with these anti-stress effects were named adaptogens; they help the human body to resist a wide range of adverse conditions such as physical, chemical, psychological, or biological stress [22]. Meanwhile, in East Asian countries, these anti-stress herbs have been regarded as herbal tonics [22]. These adaptogens or herbal tonics include but are not limited to *Panax ginseng C.A. Mey.*, *Withania somnifera*, *Ganoderma lucidum (Curtis) P. Karst*, and *Glycyrrhiza uralensis Fisch. ex Dc.* [22]. The biology of stress, such as regulation of the hypothalamic–pituitary–adrenal axis, is implicated in suicidal behavior [13]. Perceived stress is also considered an important factor mediating childhood trauma and suicide [81]. Resilience to suicidal tendencies is a set of beliefs or perceptions that buffer an individual from suicidal thoughts when faced with stressors, and can be considered a key focus of suicidal intervention [82]. Because adaptogens or herbal tonics are conceptualized as herbs that promote adaptability and resilience, the anti-stress and resilience-promoting actions of herbs in the context of suicide risk deserve investigation [83]. Given the relationship between cortisol levels and suicidal behavior [84], which is an indicator of the hypothalamic–pituitary–adrenal axis, changes in the subjects’ cortisol levels could be considered as a relevant outcome in a systematic review investigating the effectiveness of herbal medicine on suicidal behavior.

#### 3.2.4. Policy and Clinical Implications of Investigating Herbal Medicine in the Context of Suicidal Behavior

Investigating the protective effects of herbal medicine, one of the East Asian traditional medicine modalities, on suicidal behavior is related to the implementation of East Asian traditional medicine doctors in national suicide prevention policies. Clinicians with specialized training for suicide prevention are one of the prerequisites for the introduction of successful suicide prevention strategies [85,86]. In addition, the lack of quantity of medical personnel can be regarded as an obstacle to successful suicide prevention strategies. A study projecting supply and demand for doctors and nurses in Organization for Economic Cooperation and Development (OECD) countries concluded that there would be a shortage of around 400,000 doctors in 32 OECD countries in 2030 [87]. Unlike Western countries, where acupuncture and herbal medicine are regarded as types of complementary and alternative medicine, in some East Asian countries including China, Korea, Taiwan, and Japan, these East Asian traditional medicine modalities are provided by licensed clinicians in a dualized medical system [14]. Therefore, investigating the effect of herbal medicine on suicidal behavior, which mediates East Asian traditional medicine doctors and patients, can be regarded as a task to establish the justification for these medical personnel to participate in national-level suicide prevention policies. This is in line with the importance of establishing a multidisciplinary strategy for suicide prevention that is promoted today [7,8,9].

Investigating the potential impact of herbal medicine on suicidal behavior is also necessary to improve clinician practice. A substantial proportion of individuals with mental illness have been known to use herbal medicine [88,89,90]. According to the results of analyzing the National Comorbidity Survey Replication, a representative survey of American adults, the average 12-month herbal medicine use to treat mental health problems was at 3.72%, and 12.55% of individuals diagnosed with major depression used herbal medicine [88]. In addition, among patients with anxiety, the use of supplemental or herbal medicine was common at 20.52% [89]. Various lifestyle behaviors including nutrition, diet, physical activity, substance and alcohol use, and smoking are receiving increasing attention in the context of suicide risk [91]. Herbal medicine is likely to be used by individuals with mental disorders as a form of healthy behavior [92], but patient–clinician communication about the use of herbal medicine still needs to be improved [93]. Critically examining the potential effects of herbal medicine on suicidal behavior from an evidence-based medicine perspective could result in improved patient–clinician communication and improved clinician practice [93]. However, to our knowledge, the effects and safety of herbal medicine in regard to suicidal behavior have not yet been comprehensively and systematically investigated.

### 3.3. Importance of Establishing a Protocol of This Systematic Review

This systematic review protocol explicitly identifies a methodology for critically assessing the effects and safety of herbal medicines for suicidal behavior including suicidal ideation, attempts, or the complete in non-clinical or clinical populations. A protocol of systematic reviews is essential to prevent excessive duplication of this methodology and potential publication bias [94]. Suggested minimum data set for a registry of systematic reviews include research question, search strategy and date of execution of search, criteria for inclusion and exclusion, methods used to assess risk of bias, method of analysis, anticipated start date, investigators, source of funding, competing interests of authors, and date of registration [94]. The Reporting guideline, the Preferred Reporting Items for Systematic reviews and Meta-Analyses, is considered standard procedure to be followed when conducting systematic reviews [46]. In this context, protocols for systematic reviews are considered standard to adhere to the Preferred Reporting Items for Systematic reviews and Meta-Analysis Protocols 2015 statement [45]. Since the statement is not optimal, and the protocol developed is further refined in the journal’s peer-review process, this systematic review protocol prevents potential publication bias; however, the methodology of systematic reviews can be more explicit and specific [45].

## 4. Conclusions

The results of the systematic review may assist policy makers in considering herbal medicine as one of the available options in establishing national suicide prevention policies. Furthermore, it provides clinical implications for clinicians and their patients with suicidal behavior. The results of the systematic review will be presented at academic conferences and/or will be published in a peer-reviewed journal.

## Figures and Tables

**Figure 1 healthcare-11-01387-f001:**
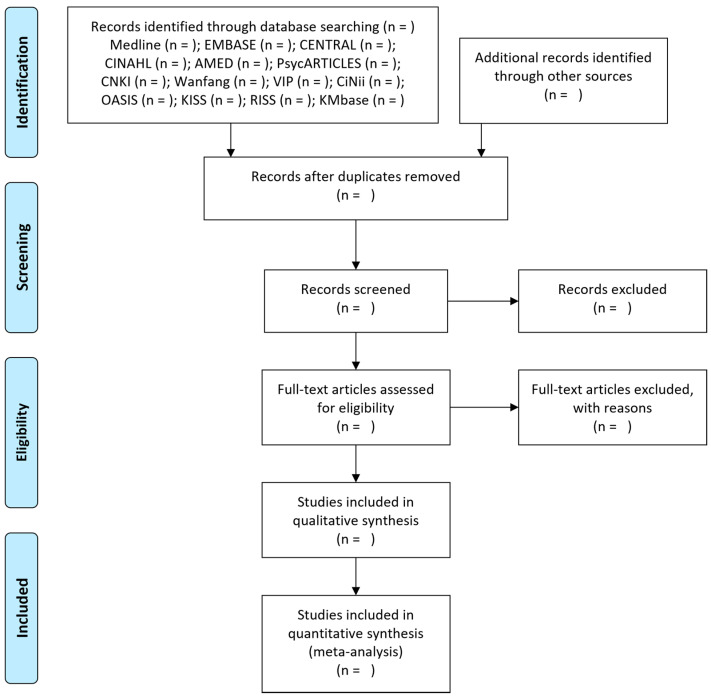
A PRISMA flow diagram showing the literature screening and selection processes. Abbreviations: AMED, Allied and Complementary Medicine Database; CENTRAL, Cochrane Central Register of Controlled Trials; CINAHL, Cumulative Index to Nursing and Allied Health Literature; CiNii, Citation Information by National Institute of Informatics; CNKI, China National Knowledge Infrastructure; EMBASE, Excerpta Medica Database; KCI, Korea Citation Index; KISS, Koreanstudies Information Service System; KMbase, Korean Medical database; OASIS, Oriental Medicine Advanced Searching Integrated System; PRISMA, Preferred Reporting Items for Systematic reviews and Meta-Analyses.

## Data Availability

The data presented in this study are available in the article and Appendix A.

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
