# Peer review of "The Effect of Herbal Medicine on Suicidal Behavior: A Protocol for Systematic Review and Meta-Analysis"

_healthcare, 2023, doi:10.3390/healthcare11101387_

Round 1

Reviewer 1 Report

Dear Editor,

The Manuscript entitled:  The efficacy of herbal medicine on suicidal behavior: A protocol for systematic review and meta-analysis has been reviewed.

This is a well-written, well-organized, and well-illustrated paper. The objectives are specific, clear, and achievable.

I have only one concern. The authors plan to put any restrictions on the types of participants. Does this affect the study's results? Isn't better to stay away from the clinical population?

I have only one concern. The authors plan to put any restrictions on the types of participants. 

Author Response

  • Response to Comments from Reviewer 1

Overall comment:

The Manuscript entitled:  The efficacy of herbal medicine on suicidal behavior: A protocol for systematic review and meta-analysis has been reviewed.

This is a well-written, well-organized, and well-illustrated paper. The objectives are specific, clear, and achievable.

Response:

Thank you for your careful review and insightful comments that have significantly enhanced our manuscript.

Comment 1:

I have only one concern. The authors plan to put any restrictions on the types of participants. Does this affect the study's results? Isn't better to stay away from the clinical population?

Response 1:           

Thank you for your comments.

There are two reasons for not limiting the type of participants in this review. First, (perhaps due to difficulties in conducting the study), 'suicidal behavior' is rarely studied compared to other diagnosed disorders. Second, suicidal behavior can also be found in non-clinical populations. (DOI: 10.1186/s13033-021-00449-z)

Therefore, we will include both the clinical and non-clinical population in this review, but strictly separate these two groups in the analysis. We have described these points in this revised manuscript.

“There will be no restrictions on the types of participants. In other words, this systematic review will include both clinical and non-clinical populations, and participants without or with suicidal behaviors will be included. This is based on the fact that suicidal behavior is difficult to study compared to other diagnosed diseases [47], and that suicidal behavior can occur not infrequently even in non-clinical populations [48,49]. Participants without suicidal behaviors for whom suicidal behavior related outcomes are not reported will be excluded. In the analysis, the clinical and non-clinical population will be strictly separated.”

(Please refer to page 5, red words)

Reviewer 2 Report

The authors of the publication, despite taking on an interesting topic, were not up to the task. The publication is only a review, there is no meta-analysis here. Furthermore, there are no results at all. Also, the authors omitted such a chapter. Section 2 very extensively describes the choice and selection of publications. The only two tables appear only in the supplementary material, and the diagram presented does not indicate the number of publications that were selected and would have met the criteria described. And the protocol mentioned in the title, which was supposed to be used in the meta-analysis and should be in the main part and on which the whole analysis of the selected publications is based, is precisely in the supplementary materials. In no way are the title and the content of the manuscript related to each other.

The entire publication is very general, does not present any specific data and it is very difficult to draw any useful conclusions from it.

A detailed list of comments follows:

1 Traditional Chinese medicine is widely used in China. However, the authors' assumption that the findings of their study will reduce suicide rates globally is an over-reaching conclusion from their literature review (line 23)

2. lines 34-42 - the paragraph is very general. It gives contradictory information and lacks examples to explain the contradictory data

3. line 47 "substance related disorder" - the term itself sounds rather strange and does not specify the type of substances that can cause mental disorders. The sentence should say that there are illnesses related to psychoactive substance abuse. Or if it refers to other compounds, the more it should be specified.

4. lines 76-79 - The authors cite a publication that deals with LGBT suicide and the 'cultural life events' described in the original publication are about, for example, family conflicts. Without specific context, the quoted paper and the whole paragraph is unreadable. Furthermore, with regard to LGBT people, how does traditional East Asian medicine affect the reduction of suicide attempts?

5 Results section missing 

6. lines 284-231 - is a continuation of the introduction. Basically the whole discussion is an expansion of the introduction.

7. Herbal medicine - is not specific only to Asian countries. The Introduction should indicate which herbs will be included in traditional East Asian medicine. Such information should also be taken into account when selecting publications.

8. lines 333-336 - again, there is a lack of detail as to which herbs were used, making it difficult to determine whether this is traditional Asian medicine

9. each paragraph of the discussion is poorly worded. The most important information appears at the end, making the work very chaotic and unreadable (for example lines 322-346)

10. lines 329-333 - there is no description of specific data, especially as the authors write: "there is preclinical and clinical evidence supporting their antidepressant effects".

11. Lines 343-346; 407-411 - the authors indicate what is to be analysed and, in fact, this is what they were supposed to analyse, as this is the topic and description in the second chapter.

12. lines351-361 - this is a very long description of suicidal behaviour that should be in the Introduction and not in the discussion

13. lines 361-363 - no data to support the effect of the compounds mentioned

14 Lines 382-396 - the description of these studies is not the subject of the publication. Such data should be presented but for herbal medicine.

15. line 379 - the lack of a case description of an increase in the risk of suicide may be due at least to the lack of studies conducted. In particular, that the authors themselves mention that suicide in Eastern culture is regarded as a weakness.

16 Lines 398-401 - the description of the study is incomplete. There is a lack of information at least on the number of patients involved in the study or what type of depression they suffered from.

17. any summary in the form of a table is missing. It is only a review of subsequent publications. It is written in an extremely chaotic manner. On the basis of which it is not possible to draw any conclusions regarding a reduction in the number of suicides as a result of herbal medicine, because, as the authors themselves state, no such studies have actually been conducted (lines 444-446).

Author Response

  • Response to Comments from Reviewer 2

Overall comment:

The authors of the publication, despite taking on an interesting topic, were not up to the task. The publication is only a review, there is no meta-analysis here. Furthermore, there are no results at all. Also, the authors omitted such a chapter. Section 2 very extensively describes the choice and selection of publications. The only two tables appear only in the supplementary material, and the diagram presented does not indicate the number of publications that were selected and would have met the criteria described. And the protocol mentioned in the title, which was supposed to be used in the meta-analysis and should be in the main part and on which the whole analysis of the selected publications is based, is precisely in the supplementary materials. In no way are the title and the content of the manuscript related to each other.

The entire publication is very general, does not present any specific data and it is very difficult to draw any useful conclusions from it.

A detailed list of comments follows:

Response:

Thank you for your careful review and insightful comments that have significantly enhanced our manuscript.

We expect you to consider that the study design of this manuscript is a protocol. In other words, as this manuscript is a protocol of a systematic review to be conducted in the future, it could not report the results of the systematic review. Nevertheless, the reason this protocol is important is to prevent undue duplication of this methodology and potential publication bias. We described this in the manuscript as follows:

“The existence of publication bias for systematic reviews has been highlighted. An international survey published in 2009 found a significant number of unpublished systematic reviews [42]. In addition, even when protocols for systematic reviews have been previously registered and/or published, discrepancies between the protocols and systematic reviews are likely to exist [43]. Since a database of systematic review protocol, the Prospective Register of Systematic Reviews, was launched over a decade ago, registration of protocols in systematic reviews has become more active, but the proportion of registered or published protocols among systematic reviews published between 2020 and 2021 was only 38% [44]. Therefore, registering a systematic review protocol, ideally, receiving peer review may contribute to securing research transparency and reducing the risk of publication bias in systematic reviews.”

(Please refer to page 4)

3.3. Importance of establishing a protocol of this systematic review

This systematic review protocol explicitly identifies a methodology for critically assessing the effects and safety of herbal medicines for suicidal behavior including suicidal ideation, attempts, or the complete in non-clinical or clinical populations. A protocol of systematic reviews is essential to prevent excessive duplication of this methodology and potential publication bias [94]. Suggested minimum data set for a registry of systematic reviews include research question, search strategy and date of execution of search, criteria for inclusion and exclusion, methods used to assess risk of bias, method of analysis, anticipated start date, investigators, source of funding, competing interests of authors, and date of registration [94]. The Reporting guideline, the Preferred Reporting Items for Systematic reviews and Meta-Analyses, is considered standard procedure to be followed when conducting systematic reviews [46]. In this context, protocols for systematic reviews are considered standard to adhere to the Preferred Reporting Items for Systematic reviews and Meta-Analysis protocols 2015 statement [45]. However, since the statement is not optimal, and the protocol developed is further refined in the journal's peer-review process, this systematic review protocol not only prevents potential publication bias, but also the methodology of systematic reviews can be more explicit and specific [45].”

(Please refer to page 11)

Regarding this response, please also refer to the comment of Reviewer 4. Except for the comment related to the study design, we tried to reflect your valuable comments in this revised manuscript as much as possible.

Comment 1:

1 Traditional Chinese medicine is widely used in China. However, the authors' assumption that the findings of their study will reduce suicide rates globally is an over-reaching conclusion from their literature review (line 23)

Response 1:           

Thank you for your comments, and we agree the reviewer's comment. Although we have given the proviso that “especially in countries that use TEAM”, the expression “globally” can be misleading, so it has been corrected in this revised version.

“Our findings will be informative for clinicians, policymakers, and researchers, aimed at reducing suicide rates, especially in countries that use TEAM.”

(Please refer to page 1, red words)

Comment 2:

  1. lines 34-42 - the paragraph is very general. It gives contradictory information and lacks examples to explain the contradictory data

Response 2:           

Thank you for your comments. We added specific country names and one research result to the pointed out sentences, as follow.

“However, the study found exceptions where more suicides than expected were observed in certain countries or regions within countries, including lower- middle-income countries (i.e, India and Iran) [5]. In addition, a repeated cross-sectional study of eight countries in four continents during the pandemic reported high rates of suicidal ideation in adults in Hong Kong [6]. This suggests that the Coronavirus disease 2019 pandemic may also have a disproportionately negative impact on suicide risk.”

(Please refer to page 1, red words)

Comment 3:

  1. line 47 "substance related disorder" - the term itself sounds rather strange and does not specify the type of substances that can cause mental disorders. The sentence should say that there are illnesses related to psychoactive substance abuse. Or if it refers to other compounds, the more it should be specified.

Response 3:           

Thank you for your comments. The reason for using the term “substance related disorder” was that the expression was used in the cited document. (doi: 10.3346/jkms.2020.35.e402) Since the study considered F10-F19 to be “substance related disorders”, we changed it to “mental and behavioral disorders due to psychoactive substance use” according to the International Classification of Diseases.

“In addition to depressive disorders, other mental health conditions including psychotic disorder, bipolar disorder, mental and behavioral disorders due to psychoactive substance use, anxiety disorders, and sleep disorders are also associated with a high risk of suicide [12].”

(Please refer to page 2, red words)

Comment 4:

  1. lines 76-79 - The authors cite a publication that deals with LGBT suicide and the 'cultural life events' described in the original publication are about, for example, family conflicts. Without specific context, the quoted paper and the whole paragraph is unreadable. Furthermore, with regard to LGBT people, how does traditional East Asian medicine affect the reduction of suicide attempts?

Response 4:           

Thank you for your comments. As the reviewer pointed out, we agree that citing the research involving LGBT people would be less directly related to this study. Therefore, the sentences have been completely modified. In this modification, some content in the Discussion section was moved to the Introduction section.

“Cultural factors are an important factor leading to consideration of traditional East Asian medicine in suicide prevention strategies. For example, Suicide prevention strategies in Asian countries should take into account Asia's unique socio-cultural context [23]. Unlike Western countries, stigma and discrimination against people with mental illness are prevalent in Asian countries [24], which is a barrier that prevents people with mental illness at risk of suicide from accessing appropriate mental health care services in those countries, including China, Japan, Korea, Hong Kong, Singapore, and Thailand [25]. Specifically, in Asian societies, mental illness is socially less accepted and often regarded as a kind of personal weakness [25]. Also in these countries, often individuals with mental illness are considered dangerous and aggressive [25]. The stigma of mental illness may itself increase the risk of suicide by contributing to an individual's social isolation and rejection [26]. Some individuals exposed to the stigma of mental disorders may be able to contact traditional East Asian medicine as a therapeutic alternative. This is because traditional East Asian medicine generally emphasizes mind-body-spirit connections and considers that treatment modalities of traditional East Asian medicine work on both the mind and body [27]. Therefore, it is highly culturally relevant to investigate the effects of treatment modalities of traditional East Asian medicine, such as acupuncture and herbal medicine, on suicidal behavior in these countries.”

(Please refer to pages 2-3, red words)

Comment 5:

5 Results section missing

Response 5:           

Thank you for your comments. As we have described above and as Reviewer 4 explained, the results section is not included because this manuscript contains a protocol.

Comment 6:

  1. lines 284-231 - is a continuation of the introduction. Basically the whole discussion is an expansion of the introduction.

Response 6:           

Thank you for your comments. In this revised manuscript, much of what was previously in the Discussion section has been moved to the Introduction section. Meanwhile, the Discussion section has been improved readability by adding appropriate subheadings.

3. Discussion

3.1. Suicide as a global health problem

Suicide is a global health problem, and has intensified in the context of the Coronavirus disease 2019 pandemic [1,2,5]. In addition, the long-term effects of the Coronavirus disease 2019 pandemic are considered a threat to an individual's mental health, including a potential increase in suicidal behavior [61]. According to a study conducted in the Australia workforce, suicidal and nonfatal suicidal behaviors carry significant socioeconomic costs [62]. In this study, the economic cost of suicide and non-fatal suicidal behavior reached $6.7 billion in 2014, calling for more action to reduce this burden [62]. In Japan, the cost of depression-related suicide was estimated at $2.5 billion in 2008 [63]. The significant socioeconomic burden associated with suicide attempts is related to the use of health services to treat the injury, the psychological and social impact of the behavior on the individual and his/her associates, and the long-term disability resulting from the injury [64]. Therefore, suicide prevention and measures must be a priority for healthcare professionals and policy makers worldwide today.

3.2. The value of investigating herbal medicine in the context of suicidal behavior

3.2.1. Its potentially beneficial effects on health conditions associated with suicide risk

A recent systematic review …

3.2.2. Complement the limitations of existing biologics

The importance of investigating …

3.2.3. Potential underlying biological mechanisms of herbal medicine for suicide risk

It is known that mental disorders …

3.2.4. Policy and clinical implications of investigating herbal medicine in the context of suicidal behavior

Investigating the protective effects …

3.3. Importance of establishing a protocol of this systematic review

This systematic review protocol …”

(Please refer to pages 8-11, red words)

Comment 7:

  1. Herbal medicine - is not specific only to Asian countries. The Introduction should indicate which herbs will be included in traditional East Asian medicine. Such information should also be taken into account when selecting publications.

Response 7:           

Thank you for your comments. Since the inclusion of studies should be described in the Methods section (i.e., types of interventions), the following was added.

“2.3.3. Types of interventions

As an intervention of interest, oral herbal medicine based on traditional East Asian medicine (i.e., traditional Chinese medicine, Korean medicine, and Kampo medicine) will be allowed. In this case, both oral herbal medicines used as monotherapy or adjuvant therapy will be allowed. However, non-oral herbal medicines such as intravenous administration, intramuscular administration (e.g., herbal injection), herbal ointment and herbal medicine retention enema will be excluded.”

(Please refer to page 5, red words)

Comment 8:

  1. lines 333-336 - again, there is a lack of detail as to which herbs were used, making it difficult to determine whether this is traditional Asian medicine

Response 8:           

Thank you for your comments. As the reviewer pointed out, the previously described sentence was unclear whether it was about an herbal medicine belonging to traditional East Asian medicine. Therefore, we have added some examples to clarify this. (The sentence has been moved to the Introduction section.)

“However, there are issues regarding the safety of herbal medicines based on traditional East Asian medicine (e.g., Panax ginseng C.A. Mey., Glycyrrhiza glabra, Angelica sinensis, and Herba Ephedrae) and their potential drug interactions in Western countries, which do not have strict regulations on herbal medicine compared to Asian countries [30].”

(Please refer to page 3, red words)

Comment 9:

  1. each paragraph of the discussion is poorly worded. The most important information appears at the end, making the work very chaotic and unreadable (for example lines 322-346)

Response 9:           

Thank you for your comments. As described above, much of what was previously in the Discussion section has been moved to the Introduction section. And the Discussion section has been improved readability by adding appropriate subheadings.

Comment 10:

  1. lines 329-333 - there is no description of specific data, especially as the authors write: "there is preclinical and clinical evidence supporting their antidepressant effects".

Response 10:         

Thank you for your comments. In this revised manuscript, the specific data has been added after the pointed sentence, as follows. (The sentence has been moved to the Introduction section.)

“For example, some herbal medicines based on traditional East Asian medicine, including Banxia houpo decoction, Chaihu shugan san, Ganmai dazao decoction, Kaixin san, Shugan jieyu capsules, Sini san, Wuling capsules, Xiaoyaosan, and Yeuju pill, have been widely used to treat depression for centuries, and there is preclinical and clinical evidence supporting their antidepressant effects [31,32]. Specifically, these herbal medicines are thought to exert antidepressant effects through multi-target and multi-path mechanisms, and potentially related terpenoids, flavonoids, xanthones, phenylpropanoids, and phenols have been identified [31]. Importantly, in a fluoxetine-adjunct, placebo-controlled, randomized controlled clinical trial on 18 patients with major depressive disorder, Yeuju pill showed a significant fast-onset antidepressant effect within 7 days [33].”

(Please refer to page 3, red words)

Comment 11:

  1. Lines 343-346; 407-411 - the authors indicate what is to be analysed and, in fact, this is what they were supposed to analyse, as this is the topic and description in the second chapter.

Response 11:         

Thank you for your comments. As we have described above and as Reviewer 4 explained, this manuscript only contains what is to be analyzed because this manuscript contains a protocol.

Comment 12:

  1. lines351-361 - this is a very long description of suicidal behaviour that should be in the Introduction and not in the discussion

Response 12:         

Thank you for your comments. As described above, much of what was previously in the Discussion section has been moved to the Introduction section. And the Discussion section has been improved readability by adding appropriate subheadings.

Comment 13:

  1. lines 361-363 - no data to support the effect of the compounds mentioned

Response 13:         

Thank you for your comments. The review article (doi: 10.3390/ph14010065) cited for the sentence described some herbal medicines or their constituents that have shown antidepressant-like activity in animal models of depression.

Since the purpose of this sentence (previous lines 361-363) was also to present the types of potentially related herbal medicines or their constituents for treating depression, only the types were described without description of the specific experimental modeling method, treatment dose, and treatment duration.

To make this clear, the sentence has been modified as follows.

“A review article found that constituents in herbal medicine that are thought to exert antidepressant-like activity through action on hormone receptors on this axis include quercetin, puerarin, pseudo-hypericin, baicalin, and Tribulus terrestris saponins in animal models of depression [79].”

(Please refer to page 9, red words)

Comment 14:

14 Lines 382-396 - the description of these studies is not the subject of the publication. Such data should be presented but for herbal medicine.

Response 14:         

Thank you for your comments. Limitations of conventional biologics are related to the value of investigating herbal medicines in this systematic review, so appropriate subheadings have been added to help understand the relevance.

3.2. The value of investigating herbal medicine in the context of suicidal behavior

3.2.2. Complement the limitations of existing biologics

The importance of investigating herbal medicines for suicidal behavior is related to the limitations of biologics in conventional medicine. Specifically, the balance of benefits and harms in the relationship between use of antidepressants and suicide risk has been controversial [71]. …”

(Please refer to pages 8-9, red words)

Comment 15:

  1. line 379 - the lack of a case description of an increase in the risk of suicide may be due at least to the lack of studies conducted. In particular, that the authors themselves mention that suicide in Eastern culture is regarded as a weakness.

Response 15:         

Thank you for your comments. We further described the association of herbal medicines with resilience and anti-stress in the context of suicidal behavior, in relation to those pointed out by the reviewer.

“Because adaptogens or herbal tonics are conceptualized as herbs that promote adaptability and resilience, the anti-stress and resilience-promoting actions of herbs in the context of suicide risk deserve investigation [83]. Given the relationship between cortisol levels and suicidal behavior [84], which is an indicator of the hypothalamic-pituitary-adrenal axis, changes in the subjects' cortisol levels could be considered as a relevant outcome in a systematic review investigating the effectiveness of herbal medicine on suicidal behavior.”

(Please refer to page 10, red words)

Comment 16:

16 Lines 398-401 - the description of the study is incomplete. There is a lack of information at least on the number of patients involved in the study or what type of depression they suffered from.

Response 16:         

Thank you for your comments. We have added to this revised manuscript the following information pointed out by the reviewer.

“For example, in a randomized placebo-controlled, pilot trial on 18 patients with major depressive disorder, compared to the group using placebo plus fluoxetine, a significant reduction in depressive symptoms was found in the group using Yeuju pill plus fluoxetine within 7 days of treatment (p < 0.05) [33].”

(Please refer to page 9, red words)

Comment 17:

  1. any summary in the form of a table is missing. It is only a review of subsequent publications. It is written in an extremely chaotic manner. On the basis of which it is not possible to draw any conclusions regarding a reduction in the number of suicides as a result of herbal medicine, because, as the authors themselves state, no such studies have actually been conducted (lines 444-446).

Response 17:         

Thank you for your comments. As we have described above and as Reviewer 4 explained, this manuscript only contains methods of this systematic review, but not conclusion, because this manuscript contains a protocol.

Reviewer 3 Report

The article sent for review is well-written, well-structured, and well-described. however, I have questions:

1. Can the authors expand on the information on the biological basis of suicide (changes in neurotransmission, receptors, gene expression)

2. Can the authors extend the information on drug interactions with natural medicine involving drug metabolizing enzymes? 

Author Response

  • Response to Comments from Reviewer 3

Overall comment:

The article sent for review is well-written, well-structured, and well-described. however, I have questions:

Response:

Thank you for your careful review and insightful comments that have significantly enhanced our manuscript.

Comment 1:

  1. Can the authors expand on the information on the biological basis of suicide (changes in neurotransmission, receptors, gene expression)

Response 1:           

Thank you for your comments. We added information on the biological basis of suicide.

“A review article examining the biological basis of suicide and suicidal behavior reported the involvement of abnormality of serotonergic mechanism, decreased serotonin metabolites, abnormalities of receptor-linked signaling mechanisms, dysregulated hypothalamic-pituitary-adrenal axis, abnormalities of neurotrophins and its receptors, and abnormalities of neuroimmune functions [13]. Specifically, important markers of suicidal behavior include decreased levels of 5-hydroxyindoleacetic acid in cerebrospinal fluid, increased platelet 5-hydroxytryptamine 2A receptors, and abnormal hypothalamic-pituitary-adrenal axis function [13]. These biological mechanisms are the premise suggesting the possibility of using biological agents for the prevention of suicidal behavior.”

(Please refer to page 2, red words)

Comment 2:

  1. Can the authors extend the information on drug interactions with natural medicine involving drug metabolizing enzymes?

Response 2:           

Thank you for your comments. We have added a description of some potential herb-drug interactions.

“Some popular herbal medicines used to improve mental health may have interactions with psychotropic drugs, for example, extract of Ginkgo biloba leaf may have effects on drug-metabolizing enzymes (e.g., inhibition of CYP450 2D6 and 3A4 [35], induction of CYP2B [36] and CYP3A4 [37]), while Panax ginseng C.A. Mey. has direct pharmacodynamic effects on the central nervous system, affecting GABAergic and/or serotonergic transmission [34].”

(Please refer to page 3, red words)

Reviewer 4 Report

The manuscript provides the fundamental steps that will be adopted to design the systematic review about the efficacy of herbal medicine on suicidal behaviour. It is well-written and provides sufficient information about the purpose of writing the manuscript. While reviewing it, I wondered why the author needs to publish the protocol separately  But I got the answer in the last paragraph of the Discussion.  quite convincing!

Additional comments

I realized that it is a protocol that is designed to conduct a systematic review and meta-analysis later on. If I take this manuscript as a "Protocol" (as mentioned in the title of the manuscript and also the journal section in which it has been submitted is named "protocol"), then it looks ok to me. As a protocol, I did not find any flaw in it. The author has explained the reason for writing this protocol in the last paragraph of the discussion (lines 447-462) very nicely.  However, if it is considered a review, it does not fulfill the criteria and does not include a compilation of any scientific data about herbal medicines impacting suicides in a systematic way. I also went through other reviewers' reports, and I definitely agree with them if I took it as a Review.  But, in the current scenario, it is a "protocol," so I disagree with other reviewers.

 Moreover, I did not find any grammatical or scientific mistakes (If I take it as a Protocol). I have already reviewed a couple of manuscripts for MDPI journals. So, I understand what would be expected from me while reviewing. But, as far as I know, I did not find any flaw. That's why I did not mention any comment for the author. However, I have one suggestion for the author right now;    1- Make the discussion precise. The discussion section is very lengthy and gives an impression of a literature review. The author can add extra explanations while writing the systematic review for which he wrote this protocol.

Author Response

  • Response to Comments from Reviewer 4

Overall comment:

The manuscript provides the fundamental steps that will be adopted to design the systematic review about the efficacy of herbal medicine on suicidal behaviour. It is well-written and provides sufficient information about the purpose of writing the manuscript. While reviewing it, I wondered why the author needs to publish the protocol separately  But I got the answer in the last paragraph of the Discussion.  quite convincing!

Response:

Thank you for your careful review and insightful comments that have significantly enhanced our manuscript.

Comment 1:

I realized that it is a protocol that is designed to conduct a systematic review and meta-analysis later on. If I take this manuscript as a "Protocol" (as mentioned in the title of the manuscript and also the journal section in which it has been submitted is named "protocol"), then it looks ok to me. As a protocol, I did not find any flaw in it. The author has explained the reason for writing this protocol in the last paragraph of the discussion (lines 447-462) very nicely.  However, if it is considered a review, it does not fulfill the criteria and does not include a compilation of any scientific data about herbal medicines impacting suicides in a systematic way. I also went through other reviewers' reports, and I definitely agree with them if I took it as a Review.  But, in the current scenario, it is a "protocol," so I disagree with other reviewers.

Response 1:           

Thank you for your accurate and detailed explanation.

Comment 2:

Moreover, I did not find any grammatical or scientific mistakes (If I take it as a Protocol). I have already reviewed a couple of manuscripts for MDPI journals. So, I understand what would be expected from me while reviewing. But, as far as I know, I did not find any flaw. That's why I did not mention any comment for the author. However, I have one suggestion for the author right now;

1- Make the discussion precise. The discussion section is very lengthy and gives an impression of a literature review. The author can add extra explanations while writing the systematic review for which he wrote this protocol.

Response 2:           

Thank you for your comments.

The reason why the content of this manuscript, especially the Discussion section, was lengthy was because the format of the protocol stipulated by this journal required more than 4000 words. However, as the reviewer points out, the Discussion section in the original manuscript gives the impression that it is not clear and is very long.

Therefore, to reflect the reviewer's comment, much of what was previously in the Discussion section has been moved to the Introduction section. And the Discussion section has been improved readability by adding appropriate subheadings.

3. Discussion

3.1. Suicide as a global health problem

Suicide is a global health problem, …

3.2. The value of investigating herbal medicine in the context of suicidal behavior

3.2.1. Its potentially beneficial effects on health conditions associated with suicide risk

A recent systematic review …

3.2.2. Complement the limitations of existing biologics

The importance of investigating …

3.2.3. Potential underlying biological mechanisms of herbal medicine for suicide risk

It is known that mental disorders …

3.2.4. Policy and clinical implications of investigating herbal medicine in the context of suicidal behavior

Investigating the protective effects …

3.3. Importance of establishing a protocol of this systematic review

This systematic review protocol …”

(Please refer to pages 8-11, red words)

Round 2

Reviewer 2 Report

The Authors responded to all comments and supplemented the manuscript with the necessary information. I have no further comments.

Reviewer 3 Report

Now this paper is ready for publication.